# A Thermally Stable Recombinant Human Fibronectin Peptide-Fused Protein (rhFN3C) for Faster Aphthous Ulcer (AU) Healing

**DOI:** 10.3390/bioengineering11010038

**Published:** 2023-12-29

**Authors:** Xiang Cai, Jiawen Zhu, Xin Luo, Guoguo Jin, Yadong Huang, Lihua Li

**Affiliations:** 1State Key Laboratory of Bioactive Molecules and Drug Gability Assessment, Jinan University, Guangzhou 510632, China; 16626412540@163.com (X.C.); 13560646034@163.com (J.Z.); luoxin@stu2023.jnu.edu.cn (X.L.); jinguoguo1709@163.com (G.J.); tydhuang@jnu.edu.cn (Y.H.); 2Institute of Biomedicine and Guangdong Provincial Key Laboratory of Bioengineering Medicine, Jinan University, Guangzhou 510632, China; 3Department of Materials Science and Engineering, Institute of Biomedical Engineering, Engineering Research Center of Artificial Organs and Materials, Jinan University, Guangzhou 510632, China; 4Biopharmaceutical R&D Center of Jinan University, Guangzhou 510632, China

**Keywords:** aphthous ulcer therapy, quick ulcer healing, recombinant fibronectin, peptide-fused protein

## Abstract

Approximately 59.4–100% of head and neck cancer patients receiving radiotherapy or radio chemotherapy suffer from aphthous ulcers (AUs), which seriously affect the subsequent treatment. At the same time, AUs are a common oral mucosal disease with a high incidence rate among the population, often accompanied by severe pain, and affect both physical and mental health. Strategies to increase the ulcer healing rate and relieve pain symptoms quickly is a long-term clinical objective. Oral mucosal discontinuity is the main histological hallmark of AUs. So, covering the inner mucosal defect with an in vitro engineered oral mucosal equivalent shows good prospects for AU alleviation. Fibronectin (FN) is a glycopeptide in the extracellular matrix and exhibits opsonic properties, aiding the phagocytosis and clearance of foreign pathogens through all stages of ulcer healing. But native FN comes from animal blood, which has potential health risks. rhFN3C was designed with multi-domains of native FN, whose core functions are the recruitment of cells and growth factors to accelerate AU healing. rhFN3C is a peptide-fused recombinant protein. The peptides are derived from the positions of 1444–1545 (FNIII10) and 1632–1901 (FNIII12–14) in human native FN. We optimized the fermentation conditions of rhFN3C in *E. coli* BL21 to enable high expression levels. rhFN3C is thermally stable and nontoxic for L929, strongly promotes the migration and adhesion of HaCaT, decreases the incidence of wound infection, and shortens the mean healing time by about 2 days compared to others (*p* < 0.01). rhFN3C may have great potential for use in the treatment of AUs. The specific methods and mechanisms of rhFN3C are yet to be investigated.

## 1. Introduction

Aphthous ulcers are typical mucosal lesions associated with many diseases, including recurrent aphthous ulcers, traumatic ulcers, and leukoaraiosis. Specifically, they are a common complication of radiotherapy and chemotherapy in cancer patients [1]. Approximately 59.4–100% of head and neck cancer patients receiving radiotherapy or radiochemotherapy suffer from an aphthous ulcer [2]. In addition to this, various diseases such as mucosal trauma, chemical irritation, and autoimmune and inflammatory ailments can lead to aphthous ulcers [3,4]. Mouth ulcers not only impair overall health but also reduce the patient’s quality of life. Aphthous ulcers are characterized by the destruction of the epithelium and the infiltration of inflammatory cells. Although ordinary aphthous ulcers are usually self-healing, this takes time and can cause unbearable long-term pain. The lamina propria is severely disrupted under this condition. Clinicians still do not have a good way to treat aphthous ulcers and suggest nothing more than oral healthcare, anti-inflammatory medications, local anesthetics, oral decontamination, and mouthwashes [5]. Therefore, it is urgent to explore new therapies for treating aphthous ulcers. Oral mucosal discontinuity is the main histological hallmark of oral ulcers. So, covering the inner mucosal defect with an in vitro engineered oral mucosal equivalent shows good prospects for use in future treatment.

The construction of oral mucosa is a relatively new field in tissue engineering. Initially, its aim is to treat and fill tissue defects caused by facial trauma or surgery for malignant lesions. During the re-establishment of human oral tissue, fibronectin (FN) is one of the first components of the extracellular matrix (ECM) to appear. It is typically found in and under keratinocytes at the anterior border of the epithelium during the healing of traumatic ulcerations, where it presumably serves as part of the ECM required for re-epithelialization [6,7]. Numerous studies have suggested that the lack of FN is a contributing factor to the delayed healing of ulcers in patients [8,9]. In other chronic wounds, such as venous ulcers, diabetic ulcers, and pressure ulcers [10], fibronectin fragments are present in the wound fluid, indicating proteolytic digestion of the ECM, which is a possibility in these lesions [11].

As a key factor causing delayed diabetic wound healing [12,13], FN is a large-molecular-weight glycoprotein found abundantly in the ECM [14,15]. FN originating from plasma of the human body contributes to phagocytosis and the clearance of foreign pathogens by neutrophils and macrophages and releases a variety of growth factors and cytokines that bind to keratinocytes, fibroblasts, and endothelial cells, during which a large number of cells migrate to the surfaces of wounds to promote wound healing [16]. The protein structure of FN has been assessed. Each monomer of the FN consists of three types of repeat units (called FN repeats): type I, type II, and type III. Type III repeats are approximately 90 residues long, without any disulfide bonds, and these units sequentially form structural and functional domains that mediate binding to cell surface receptors or other ECM molecules [17,18]. Among them, the most widely known is the RGD sequence located in FNIII10, which contains the sequence Arg–Gly–Asp. FNIII10 not only regulates cell adhesion but also plays an important role in cell survival, proliferation, and differentiation. It is mostly used in tissue engineering for the modification of various polymers and inorganic materials to enhance their biocompatibility and cell adhesion [18,19]. Sana et al. used RGD and fibronectin-immobilized chitosan scaffolds to piggyback human dental pulp stem cells (hDPSCs) and found the scaffolds can significantly promote cell migration, proliferation, and differentiation [20]. Our team [21] combined FNIII10 and CC (human collagen-like protein) proteins to obtain FCP. The ECM-mimetic FCP provides a microenvironment that supports stem cell localization, proliferation, differentiation, and stemness maintenance. However, its monofunctionality limits its application in oral ulcer healing.

Growth factors are strongly involved in the wound-healing process [22]. FNIII12–14 contains the heparin-binding domain II and a highly intermixed growth factor-binding domain, which has been shown to exhibit strong affinity for vascular endothelial growth factor (VEGF), platelet-derived growth factor (PDGF), fibroblast growth factor (FGF), and transforming growth factor-beta (TGF-β) [23]. To achieve the best efficacy of FN-based fused protein, this peptide-fused protein should contain both the mentioned functional domains: one promoting cell adhesion and one promoting growth-factor binding. Their advantages can be exploited to make up for deficiencies, thus yielding a win–win result. On the other hand, fibronectin generally has low solubility, leading to post-processing difficulty. The recombinant expression of truncated FN via new techniques of molecular biology can provide soluble FN. We have designed a truncated FN fragment, aspiring to the effects of solubility and pro-stabilization. In addition to this, this fragment is involved in interactions with α5β1 integrins and growth factor receptors that enhance the signaling and structural effects [24].

In light of all of the above, based on the structural and functional characteristics of two fragments, FNIII10 and FNIII12–14, of fibronectin were prepared and combined in this study to obtain recombinant human fibronectin truncated peptide. Here, a novel recombinant human fibronectin truncated peptide (rhFN3C) with the ability to promote cell proliferation, migration, and adhesion was created using genetic engineering. rhFN3C promoted cell migration more effectively than its structural domain fragments FNIII10 and FNIII12–14. rhFN3C is heat stable, non-toxic to L929, promotes HaCat migration and adhesion, reduces the incidence of wound infection, and shortens the wound healing time. However, the underlying mechanisms remain unclear and require further investigation.

## 2. Materials and Methods

### 2.1. Instruments and Materials

#### 2.1.1. Reagents and Solvents

Vectors pET-20b (Invitrogen, Guangzhou, China) and *E. coli DH5α* (Invitrogen, Guangzhou, China, ATCC^®^ BAA-3219™) were used for cloning and heterologous expression. The PCR purification kits, gel extraction kits, and micropreparation kits were purchased from Tiangen (Beijing, China). The gene encoding rhFN3C was cloned into the pET-20b vector to produce the recombinant plasmid pET20b-rhFN3C. The positive clone was screened for ampicillin resistance and transformed into *E. coli BL21 (DE3)* and *pLysS* (Invitrogen, Guangzhou, China, ATCC^®^ BAA-1025™). For induction by Isopropyl-beta-D-thiogalactopyranoside (IPTG), the recombinant proteins were examined and identified by SDS- polyacrylamide gel electrophoresis (SDS-PAGE) and Western blot (WB) analysis, and their purity was determined by HPLC. The high-expressing strains were selected for a study of the fermentation process in a 15 L fermenter pilot. Fermentation was performed by inoculating the seed solution at 10% and supplementing with carbon and nitrogen sources to promote growth; induction by IPTG was performed during the logarithmic growth phase. In addition, rhFN3C protein was purified by affinity chromatography on a Ni-Sepharose 6 Fast Flow column in combination with Sephadex G-25 gel filtration.

#### 2.1.2. Cell Cultures

L-929 cells (ATCC) were purchased from the Chinese Academy of Sciences (Shanghai, China) and cultured in Dulbecco’s modified Eagle’s medium (DMEM) containing 10% fetal bovine serum (FBS) (Gibco, Grand Island, NY, USA) and 1% penicillin/streptomycin (100 units/mL) in a humidified atmosphere at 37 °C with 5% CO_2_. 

The Chinese Academy of Sciences (Shanghai, China) provided the BALB/c 3T3 cells (ATCC), which were then cultured in RPMI 1640 supplemented with 10% fetal bovine serum (FBS) (Gibco, Grand Island, NY, USA). 

The Chinese Academy of Sciences sold HaCaT cells (ATCC), which were cultivated in DMEM enhanced with 10% FBS. Every flask and plate used for cell culture was bought from Corning (Corning, NY, USA).

The culture medium was changed every 2 days, and cells were dissociated at 80% confluence using trypsin. 

### 2.2. Methods

#### 2.2.1. Expression of Recombinant Fibronectin Peptide-Fused Protein in *E. coli*

The DNA fragments encoding the recombinant fibronectin peptide-fused protein rhFN3C (ID: KAI2526825.1, FNIII10: aa1444 to aa1545 and FNIII12–14: aa1633 to aa1901) were transferred to BL21(DE3) and BL21 pLysS and the expressing bacteria with the sequence were named BL21(DE3)/pET20b-rhFN3C and BL21 pLysS/pET20b-rhFN3C, respectively. The sequences of the FNIII10 and FNIII12–14 gene fragments used in the latter experiments were amplified using PCR. Subsequent double enzymatic cleavage of the FNIII10 and FNIII12–14 fragments was performed with the enzymes EcoR I and Xho I, respectively. Double enzymatic cleavage of the pET20b plasmid was performed with the enzymes Nde I and Xho I, both of which were reacted in a water bath for 1 h at 37 °C. After ligation of the gene fragments of FNIII10 and FNIII12–14 with the digestion products of the linearized vector pET20b, the above ligated products were transformed into *E. coli DH5α*, and the recombinant plasmids were obtained by screening for positive clones [25].

#### 2.2.2. Screening and Expression of Recombinant Fibronectin Peptide-Fused Protein

Three BL21(DE3)/pLysS-positive single colonies containing the recombinant plasmid pET20b-rhFN3C were selected separately and inoculated into liquid LB medium containing Amp+ before being cultured overnight at 37 °C. The seed solution was inoculated at a volume ratio of 1:100 into fresh LB liquid medium and incubated at 37 °C with constant shaking until OD_600_ = 0.6. IPTG was added at a final concentration of 1.0 mM and the culture was induced for 4 h. Bacteria were collected by centrifugation at 4500 rpm for 15 min and expression was identified using SDS-PAGE and WB.

#### 2.2.3. Purification and Characterization of Recombinant Fibronectin Peptide-Fused Protein

A high-pressure homogenizer was precooled to 4 °C with 300 mL of equilibrium solution. A resuspension of 30 g bacteria was subjected to uniform atmospheric pressure once, then crushed at 1000 Pa for 3 times until the bacterial suspension became clarified and transparent; this was then collected into a centrifuge tube, and subjected to 15,000 rpm centrifugation for 30 min at 4 °C to obtain the supernatant sample. We obtained solutions at different stages of the process of purification of the nickel column and identified them using SDS-PAGE.

#### 2.2.4. Thermal Stability of rhFN3C

The thermal stability of rhFN3C was evaluated with reference to the Chinese Pharmacopoeia (2020 edition). rhFN3C and bovine serum albumin (BSA, Macklin, Shanghai, China) were kept in a water bath at 55 °C for 72 h.

During the test period, samples for protein concentration testing were taken at 0, 1, 3, 5, 57, 12, 24, 48, and 72 h. The assays were performed according to instructions of the BCA protein concentration assay kit (ThermoFisher Scientific, Waltham, MA, USA).

#### 2.2.5. rhFN3C Cytotoxicity against L929 Cells

Cell Preparation: A concentrated suspension of L929 cells at a concentration of 1 × 10^5^ cells/mL was prepared in growth medium. The cell suspension was inoculated into 96-well plates at 100 μL per well, and all wells were incubated at 37 °C with 5% CO_2_ for 24 h until the cells reached 70–80% fusion.

The 96-well plate with growing cells was removed, and rhFN3C at concentrations of 62.5 μM, 125 μM, 250 μM, and 500 μM, as well as the control solution, were added to different wells. Repeat this procedure for 5 wells for each concentration. Incubation was continued for 24 h at 37 °C in 5% CO_2_. The original growth solution was discarded and 50 μL of 1 mg/mL MTT solution was added to each well. The MTT solution was removed from the 96-well plate and 100 μL of isopropanol was added to each well. The plates were stirred for 10 min with an oscillator. The optical density (OD) of each well was measured at 570 nm using an enzyme immunoassay analyzer. Absorbance values were recorded for each well, the average of the five wells was calculated, and the cell proliferation rate for each group was calculated using the formula [26].
$$Percent\;cytotoxicity=\frac{\left(A_{Experimental\;value}-A_{negative\;control}\right)}{\left(A_{positive\;control}-A_{negative\;control}\right)\;}\times 100\%$$

#### 2.2.6. Scratch Test to Detect the Pro-Migratory Effect of rhFN3C on HaCaT Cells

Cell preparation: HaCaT cells were cultured in complete growth medium (10% FBS, H-DMEM) at 37 °C, 5% CO_2_, and cell density was controlled at 1–5 × 10^6^ cells/mL.

The well-grown HaCaT cells were taken and digested for 5 min by adding 1.5 mL of trypsin containing 0.25% EDTA. After digestion, the cells were counted and diluted into a cell suspension of 2 × 10^5^ cells/mL. A total of 2 mL of cell suspension was inoculated into each well of a 6-well plate. The dishes were transferred to a 37 °C, 5% CO_2_ cell culture incubator and incubated for 48 h until the cells were fully grown.

Five parallel lines were drawn in each well with a 200 μL pipette tip. The scraped cells were then washed three times with sterile PBS. Serum-free H-DMEM medium containing 125 μM rhFN3C was used for the experimental group, and serum-free H-DMEM medium was used for the control group. Control 1 used serum-free basal medium containing 125 μM FN10 (i.e., FNIII10). Control 2 used serum-free basal medium containing 125 μM FN12–14 (i.e., FNIII12–14). The healing of the scratches was observed and photographed at 0, 24, and 48 h using an inverted microscope. The area of the scratched area was analyzed using ImageJ 1.52 software.

#### 2.2.7. Crystal Violet Cell Adhesion Assay to Test the Promoting Adhesion Effect of rhFN3C on BALB/c 3T3 and HaCaT Cells

On the day before the experiment, we coated a 24-well plate with serum-free RPMI 1640 medium containing 125 μM rhFN3C. Two control groups were established, one with serum-free basal medium containing 125 μM FN10 and the other with serum-free basal medium containing 125 μM FN12–14. The control group consisted of serum-free RPMI 1640 medium. Each group consisted of 3 replicate wells. These were incubated at 4 °C overnight.

As previously described [27], before the experiment, we discarded the supernatant and washed the wells 3 times with sterile PBS. We digested the effectively growing 3T3 cells with trypsin and resuspended the cells in serum-free RPMI 1640 medium. We diluted the cells to a concentration of 2 × 10^4^ cells/mL, inoculated 500 μL of the cell suspension into each well of the 24-well plate, and incubated them at 37 °C for 3 h. We discarded the culture supernatant, washed the wells 3 times with sterile PBS, fixed them with a 4% paraformaldehyde solution for 30 min, washed them 3 times with PBS, and added 500 μL of a 1% crystal violet staining solution to each well for 30 min. This was washed 3 times with PBS; the samples were then observed under an inverted microscope, after which we randomly selected and recorded 5 image fields for each group. We analyzed the number of adherent cells using ImageJ 1.52 software. The growth medium for HaCaT cells was H-DMEM, and the experimental steps were the same as described above.

#### 2.2.8. Therapeutic Effect of rhFN3C on Aphthous Ulcer Healing

All SD rats used in this study (60 days after birth) were purchased from Guangdong Animal Center (No. 44007200069979) and kept in an appropriately controlled environment. The experimental protocol used in this study was approved by the Institutional Animal Care and Use Committee of Jinan University (approval number 20220302-18). All experiments were conducted in accordance with the Chinese Guidelines for the Care and Use of Animals, and were approved by the Animal Ethics Committee of the Chinese Academy of Medical Sciences.

Fifty SD male rats aged 8–10 weeks were selected and housed in the laboratory for three days. Anesthesia was administered by intraperitoneal injection of 2% sodium pentobarbital (30 mg/kg); 6 mm diameter filter paper was immersed in 70% acetic acid for 2 min, and then placed on both sides of the oral mucosa for 1 day [28]. After modeling, photographs were obtained and the formation of aphthous ulcers was evaluated. 

The ulcer model rats were randomly divided into 5 groups of 10 rats each: control, vehicle, collagen (COL1, 400 μg/mL), rhFN3C 200 (200 μg/mL), and rhFN3C 400 (400 μg/mL). The day the ulcer model was created was recorded as day 0, and the drugs (COL1, rhFN3C) were injected on days 1, 3, 5, and 7 using the 5-point injection method (5-point injection, i.e., the injection point is located in the center of the ulcer). The vehicle group was injected with an equal volume of PBS. The conditions and methods of injection were compatible with the drug.

We photographed the aphthous ulcers on days 1, 3, 5, 7, and 9 to observe their healing. On the ninth day of modeling, the rats were slain. The buccal mucosa of the ulcerated side and the healthy side (the buccal mucosa of the ulcerated side contained at least 1 mm of normal mucosal tissue) were removed, fixed, embedded, and sequentially sliced at a thickness of 4 mm to observe the healing process of the ulcers.

#### 2.2.9. Statistical Analysis

Data are presented as mean ± standard deviation (SD) based on at least three independent tests. Data analysis was performed using GraphPad Prism 9 (GraphPad Inc., Jolla, CA, USA). In the case of different groups, a one-way ANOVA was performed, followed by Tukey’s Honestly Significant Difference (HSD) comparison test. The statistical significance of the results was set at *p* < 0.05.

## 3. Results

### 3.1. Construction, Expression, and Identification of rhFN3C

The DNA fragment encoding the recombinant human fibronectin peptide-fused protein was transferred into BL21(DE3) and BL21 pLysSand. The expressing bacteria with the sequence were named BL21(DE3)/pET20b-rhFN3C and BL21 pLysS/pET20b-rhFN3C, respectively. Gene fragments FNIII10 and FNIII12–14 were amplified using PCR to generate the sequences used in subsequent experiments. As shown in Figure 1A, the recombinant expression vector pET20b-rhFN3C was successfully constructed, and BL21 (DE3)/rhFN3C and pLysS/rhFN3C were obtained. As shown in Figure 1C,D, the identification results showed that rhFN3C was successfully expressed in BL21 (DE3) and pLysS. As shown in Figure 1B, the results indicated that induction was induced at 30 °C for 6 h. The highest relative percentage of total bacterial content was expressed at an IPTG concentration of 0.5 mM, and 0.5 mM of IPTG at 30 °C for 6 h was subsequently used as the induction condition. As shown in Figure 1E, recombinant protein rhFN3C with high purity was obtained except for gelatin ion exchange chromatography. Using nickel ion affinity chromatography, the recombinant protein rhFN3C was obtained with a purity of >90%, as shown in Figure 1F,G.

### 3.2. Thermal Stability of rhFN3C

The results (Figure 2) of the thermal stability test of rhFN3C show that the protein concentration of rhFN3C under the test conditions showed less variation during the test period. The variation ranges of the concentration were less than 5%, indicating that rhFN3C had excellent thermal stability. 

### 3.3. Cell Viability Assessment after Exposure to rhFN3C

#### 3.3.1. rhFN3C Cytotoxicity on L929 Cells

Figure 3A demonstrates that, within the experimental concentration range, rhFN3C had no significant proliferative effect on L929 cells. Furthermore, even at higher concentrations, it had no significant toxic effects.

#### 3.3.2. Scratch Test to Detect the Promigratory Effect of rhFN3C on HaCaT Cells

The cell migration of each group was observed and photographed under a microscope every 24 h after administration, and the results are shown in Figure 3B. The healing rate between groups was analyzed using ImageJ. Compared to the control group, there were significant differences in the cell migration healing area in the rhFN3C group after 24 and 48 h of administration (*p* < 0.001), and the wound was completely healed at 48 h. At 24 h and 48 h, the rhFN3C group showed better promotion of cell migration compared to the structural domain fragment FN10 and FN12–14 groups (*p* < 0.05).

#### 3.3.3. Crystal Violet Cell Adhesion Assay to Test the Promoting Adhesion Effect of rhFN3C on BALB/c 3T3 and HaCaT Cells

The ability of rhFN3C to promote cell adhesion in HaCaT and BALB/c 3T3 cells was evaluated using a crystal violet cell adhesion experiment. As shown in Figure 3C, microscopic observation revealed that both HaCaT and BALB/c 3T3 cells in the control group exhibited less cell adhesion and incomplete spreading, while the protein-treated groups showed more cell adhesion and elongated cell morphology. Five fields were randomly selected from each group for cell counting using ImageJ. The crystal violet adhesion results of HaCaT cells indicate that there were statistically significant differences in cell adhesion between the FN10 group and the control group (*p* < 0.001), as well as between the FN12–14 group and the control group (*p* < 0.001). Among them, the rhFN3C group had the highest number of adherent cells—significantly higher than the control group (*p* < 0.0001) and the FN10 group (*p* < 0.05), while the crystal violet adhesion results of BALB/c 3T3 cells show statistically significant differences in cell adhesion between the FN10 group and the control group (*p* < 0.05), as well as between the FN12–14 group and the control group (*p* < 0.001). Among them, the rhFN3C group showed the highest number of adherent cells, significantly higher than the control group (*p* < 0.001) and the FN10 group (*p* < 0.05).

### 3.4. rhFN3C Promoted Aphthous Ulcer Healing in SD Rats

Figure 4B depicts aphthous ulcers in rats on days 1, 3, 5, 7, and 9. On day 5, the ulcer area was reduced significantly in the collagen group, the rhFN3C 200 group, and the rhFN3C 400 group. Figure 4C depicts the daily dynamics of the ulcer area recovery percentage. On Day 1, following the formation of oral mucosal ulcers in rats, we saw no recovery of ulcer area in any of the groups. Comparing the wounds of each group revealed that the protein-supplemented group had a greater pro-healing effect on the aphthous ulcers of rats, with nearly 80% of the ulcers healing by Day 7, whereas the wounds of the other groups remained more severe at Day 9. rhFN3C delivery had a better pro-healing effect on the aphthous ulcers of rats, with individual animals in the rhFN3C 400 group almost completely healed by Day 9. 

Figure 5A,B (40× and 200× magnification) illustrates the histopathologic differences between the different groups. Figure 5A demonstrates that hematoxylin–eosin staining revealed ulcerated mucosa with epithelial layer ruptures and losses of basement membrane in all five groups after the first day of administration (Day 2). On Day 3, inflammatory cell infiltration was lower in the rhFN3C group than in the ulcerated group. On Day 5, there was a thin layer of epithelial formation in the ulcer in the rhFN3C group, but no epithelial tissue repair was observed in the other groups. On Day 9, the structural tissue of the model group was destroyed by inflammatory infiltration; the upper epidermis of the blank matrix group began to form; the COL1 and rhFN3C groups formed upper epidermis and basal layers, the collagen arrangement was improved, and the wound was effectively repaired.

Masson staining revealed that collagen fiber repair had begun in the rhFN3C group on Day 3 after modeling, whereas the ulcer group had not yet undergone obvious collagen fiber repair, and collagen formed in the rhFN3C group on Day 7 after modeling was coarser and more regularly arranged than in the ulcer group. On Day 9, the rhFN3C group demonstrated superior structural organization and increased the collagen content (Figure 5B).

## 4. Discussion

The clinical picture of aphthous ulcers is characterized by multiple, recurrent, small, round, or egg-shaped ulcers with circumscribed edges and erythematous halos of varying sizes [29]. During healing, the participation of substances such as fibronectin, elastin, type I collagen, and type III collagen is required. FN, as an important component, can effectively promote healing.

The wound healing of aphthous ulcers is often separated into four overlapping phases, i.e., hemostasis, inflammation, proliferation, and remodeling [30]. FN is critical throughout the entire process of wound healing. Fibronectin is abundantly deposited in the ECM during wound healing to improve cellular functions, promote angiogenesis, reduce inflammation, and modulate collagen deposition and degradation [31,32]. As an essential element, fibronectin can also improve healing quality by preventing scar formation through the degradation of collagen [33,34]. Various isoforms or FN fragments have been designed and shown to be more effective than the full-length FN [35] at selectively promoting wound healing. Daniel C. Roy et al. inserted the integrin-binding Arg–Gly–Asp (RGD) of FNIII10 into FNIII1H to generate a chimeric fibronectin fragment (FNIII1HRGD). FNIII1HRGD increased the granulation tissue thickness and accelerated wound closure in total excision wound experiments on genetically diabetic mice [36].

Reepithelialization and reconstruction of the epithelial layer were faster in our study after administration of rhFN3C. Keratinocytes and fibroblasts played an extremely important role in this process. Therefore, we focused on keratinocytes (HaCaT) and fibroblasts (BALB/c 3T3) to investigate the bioeffects of rhFN3C. We observed that rhFN3C had a positive effect on the migration and adhesion of HaCaT and BALB/c 3T3 in vitro, which is consistent with the results in vivo. In rats with aphthous ulcers, the healing time was not significantly different between the COL1 and rhFN3C groups. This may be associated with the main functional domains of rhFN3C: FNIII10 and FNIII12–14. The RGD sequence in FNIII10 not only regulates cell adhesion, but also plays an important role in cell recruitment, survival, proliferation, and differentiation. In addition, FNIII12–14 contains heparin-binding domain II and highly promiscuous growth factor-binding domains. The sequences of rhFN3C come from plasma and show an EGF-like bioactivity that benefits the healing of ulcers. In addition, normal oral epithelium forms finger-like projections called nail protrusions, which increase the adhesion area between the epithelial layer and the underlying layer of the oral mucosa, maintaining the stability of the oral epithelium. FN, as a ligand for integrin, is present in the keratinocytes at the leading edges of healing traumatic ulcers and their surroundings. rhFN3C may be an important molecule for the formation of nail protrusions and the repair of the oral epithelium. However, due to time constraints, this aspect was not explored in this study. Further experimental investigations will be conducted in the future.

Compared with collagen, the advantage of FN is that it can directly communicate with cells and regulate cell fate. Many studies have demonstrated that fibronectin is more expressed in oral tissue than in skin [37,38,39]. Fibronectin is abundant in the dermis and dermal–epidermal basement membrane regions during wound healing [33]. It assembles into a complex fibrillar network on the oral mucosal surface, which is vital for establishing and maintaining tissue architecture and for regulating cellular processes, including adhesion, spreading, proliferation, migration, and apoptosis [38,40,41]. FN can immobilize collagen and synergize collagen to exploit its advantages fully [42,43]. Collagen has been widely used in the bioengineering and pharmaceutical industry for many years [44], so we think FN would show broad prospects for use as a supplement of collagen.

## 5. Conclusions

The recombinant plasmid pET20b-rhFN3C was constructed on the basis of original plasmids in the laboratory by enzyme digestion and linking and was transformed into the expression systems of *E. coli BL21 (DE3)* and pLysS for strain-screening and the optimization of expression conditions. This enabled us to obtain engineered strains with high expressions of rhFN3C. The best purification methods were selected using nickel column affinity chromatography, heparin affinity chromatography, SP ion exchange chromatography, and CM ion exchange chromatography. The protein purity of G25 gel filtration was determined using HPLC after desalting. The stability of rhFN3C was determined by accelerated thermal stability tests. In this study, L929, HaCaT, and BALB/c 3T3 were used as model cells. The effects of rhFN3C on cell proliferation, migration, and adhesion were studied using MTT, cell migration assays, and cell adhesion crystal violet staining. Furthermore, in vivo experiments in SD rats assessed the efficacy of rhFN3C in boosting the aphthous ulcer-healing process. 

## Figures and Tables

**Figure 1 bioengineering-11-00038-f001:**
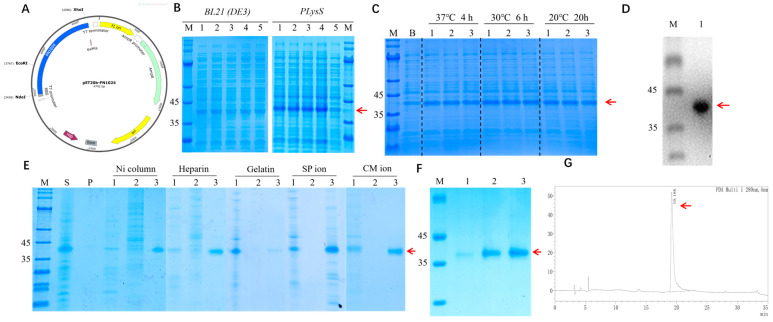
Construction and expression of rhFN3C. (**A**) Recombinant plasmid map of pET20b-rhFN3C. (**B**) SDS-PAGE analysis of the expression level of rhFN3C in different expression strains. M: protein marker. Lanes 1–4: induced cells. Lane 5: uninduced cells. (**C**) SDS-PAGE analysis of the effects of different temperatures, induction times, and IPTG concentrations on protein expression. M: protein marker. B: uninduced. Lanes 1–3: 0.25, 0.5, and 1.0 mM IPTG, respectively. (**D**) WB identification of recombinant protein rhFN3C. M: protein marker. (**E**) Screening and optimization of the rhFN3C purification process. S: cell lysate. P: cell pellet. M: protein marker. Lanes 1–3: flow-through, wash, and elution, respectively. (**F**) SDS-PAGE analysis of protein purity in the eluate from G25 gel filtration chromatography. M: protein marker. Lanes 1–3: elution solutions collected at different times. (**G**) HPLC analysis of protein purity of rhFN3C. Red arrows (**A**–**F**): Destination bands for rhFN3C. Red arrow (**G**): The characteristic peaks of rhFN3C.

**Figure 2 bioengineering-11-00038-f002:**
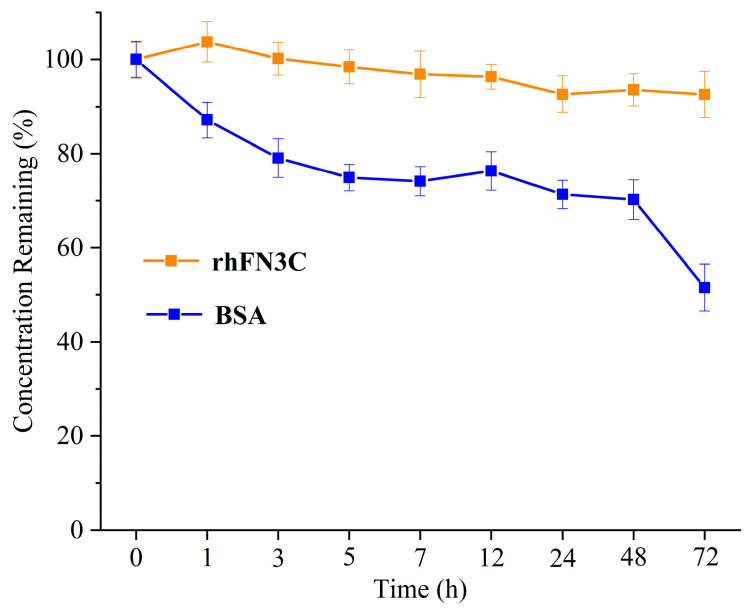
The line chart of remaining percentage comparisons of rhFN3C and BSA.

**Figure 3 bioengineering-11-00038-f003:**
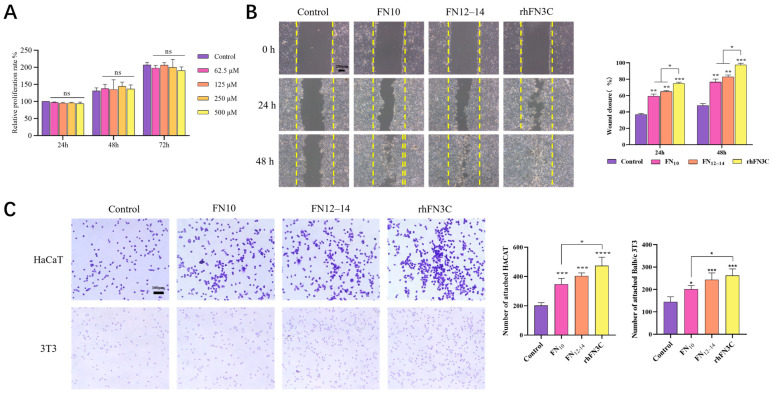
Performance comparison of rhFN3C with FN10 and FN12–14. (**A**) The MTT method detects the proliferative effects of various concentrations of rhFN3C on L929 cells after 24 h, 48 h, and 72 h of culture. Note: n = 3, ns = No Significant. (**B**) The migration and healing rate of HaCaT cells induced by rhFN3C. Note: n = 3; *, *p* < 0.05; **, *p* < 0.01; ***; and *p* < 0.001 (scale bar = 200 μm). (**C**) Detection of the adhesive effect of rhFN3C on HaCaT and BALB/c 3T3 cells. Note: n = 3; *, *p* < 0.05; ***, *p* < 0.001; ****, and *p* < 0.0001 (scale bar = 100 μm).

**Figure 4 bioengineering-11-00038-f004:**
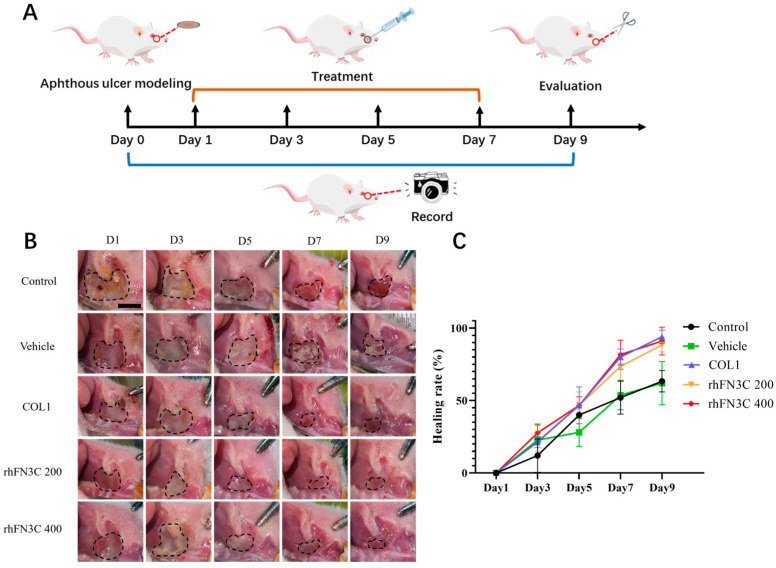
Experimental study on the healing of aphthous ulcers in rats treated with different concentrations of rhFN3C. (**A**) Schematic illustration of the experimental procedure. (**B**) Condition of the wound in rats with aphthous ulcers on days 1, 3, 5, 7, and 9 (scale bar = 0.5 mm). Note: The black circle = Unhealed areas. (**C**) The healing rate of the wound in rats with aphthous ulcers. n = 6, means ± SD.

**Figure 5 bioengineering-11-00038-f005:**
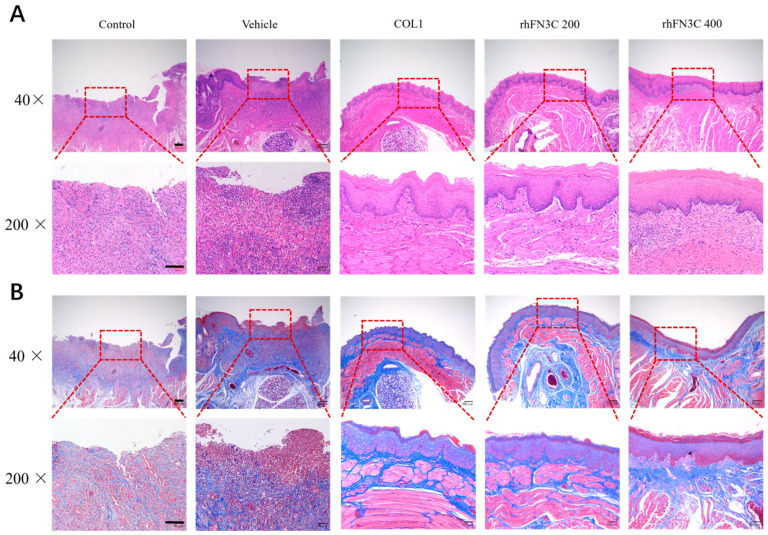
Histological observations. (**A**) Transmission light of H&E-stained images of defect sites treated with rhFN3C on Day 9 after surgery (40×: scale bar = 200 μm; 200×: scale bar = 100 μm). (**B**) Transmitted light pictures of the Masson’s trichrome-stained parts of the defects treated with rhFN3C at 9 days post-surgery, indicating collagen deposition (40×: scale bar = 200 μm; 200×: scale bar = 100 μm).

## Data Availability

The raw data supporting the conclusion of this article will be made available by the authors, without undue reservation.

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
