# Peer review of "A Thermally Stable Recombinant Human Fibronectin Peptide-Fused Protein (rhFN3C) for Faster Aphthous Ulcer (AU) Healing"

_bioengineering, 2023, doi:10.3390/bioengineering11010038_

Round 1

Reviewer 1 Report (Previous Reviewer 3)

Comments and Suggestions for Authors

The manuscript is acceptable for publication in this format.

Author Response

Reviewer 2 Report (Previous Reviewer 2)

Comments and Suggestions for Authors

I am satisified with the revised version

Author Response

Reviewer 3 Report (Previous Reviewer 1)

Comments and Suggestions for Authors

One of the major comments concerns the term "peptide": usually peptide refers to 25-30 residues, 60-80 residues as polypeptide and higher length sequences are proteins!!!! The authors should deeply revise these concepts in the manuscript.

Again the authors wrote: "The peptides are derived from the position of 1444-1545 (FNIII10) and 1632-1901 (FNIII12- 23 14) in human native FN" a part the English which is not correct, but they never report on the previous studies which pointed out the importance of protein region

The abstract is still unclear in my opinion

Round 2

Reviewer 3 Report (Previous Reviewer 1)

Comments and Suggestions for Authors

the authors finally address my issues

Author Response

This manuscript is a resubmission of an earlier submission. The following is a list of the peer review reports and author responses from that submission.

Round 1

Reviewer 1 Report

Comments and Suggestions for Authors

The great problem of this manuscript is its scientific assumption why use truncated peptide (rhFN3C)? Which is the term truncated? How long is this peptide/ why does it need to be expressed and not synthesized?

This part is completely absent in the manuscript and makes very difficult the reading of the manuscript

Other issues concern the organization of the results which often include details of mat and methods.

I advice the authors to deeply revise the manuscript pointing out the scientific basis of their study and the main results. In the present form is very difficult to appreciate these points

Comments on the Quality of English Language

medium quality

Reviewer 2 Report

Comments and Suggestions for Authors

The authors optimizes the fermentable conditions of rhFN3C in E. coli BL21 for high level expression. I have following queries regarding this manuscript

1)                What are the symptoms associated with aphthous ulcer?

2)                How fibronectin plays an important role in healing of ulcer?

3)                What are the stages of ulcer healing by fibronectin ?

4)                what is the mechanism of action of rhFN3c?

5)                What methods of cell culture are used in the given article?

6)                Prove the thermo stability of rhFN3c?

7)                How cytotoxicity of rhFN3c was assayed in the given article?

8)                What are the therapeutic outcomes of rhFN3c ?

Comments on the Quality of English Language

Some minor changes needed

Reviewer 3 Report

Comments and Suggestions for Authors

The manuscript presents an interesting and novel approach to aphthous ulcer healing. The authors have not clearly demonstrated the significance of their work. However, some comments need to be made in this report.

-Arrangement of the keywords according to MeSH terms and rearrangement according to the English alphabet.

-The aim(s) of the manuscript needs to be clearer.

-The authors have not clearly demonstrated the significance of their work, and it needs to be discussed in another way according to the aim(s) and the results.

Round 2

Reviewer 1 Report

Comments and Suggestions for Authors

The authors did not satisfy my questions.1)I asked for the scientific reason for truncated peptides and no explanation was given 2)I asked for a technical reason for expression instead of the chemical synthesis of single peptides but the authors referred to the entire protein... it is a non-sense! The peptide sequences are not reported in any figure. The authors refer to these peptides once as a single sequence and other times as a pool of sequences also in the answer. The manuscript in my opinion was not improved in its scientific message and thus needs to be rejected

Comments on the Quality of English Language

Very bad English also in the answer letter
